# Inhibition of Respiratory Syncytial Virus Infection by Small Non-Coding RNA Fragments

**DOI:** 10.3390/ijms23115990

**Published:** 2022-05-26

**Authors:** Sandra Axberg Pålsson, Vaishnovi Sekar, Claudia Kutter, Marc R. Friedländer, Anna-Lena Spetz

**Affiliations:** 1Department of Molecular Biosciences, The Wenner-Gren Institute, Stockholm University, 10691 Stockholm, Sweden; 2Science for Life Laboratory, Department of Molecular Biosciences, The Wenner-Gren Institute, Stockholm University, 10691 Stockholm, Sweden; vaishnovi.sekar@scilifelab.se (V.S.); marc.friedlander@scilifelab.se (M.R.F.); 3Science for Life Laboratory, Department of Microbiology, Tumor and Cell Biology, Karolinska Institute, 17177 Stockholm, Sweden; claudia.kutter@ki.se

**Keywords:** Respiratory Syncytial Virus (RSV), sncRNAs, tRNA, YRNA, rRNA

## Abstract

Respiratory syncytial virus (RSV) causes acute lower respiratory tract infection in infants, immunocompromised individuals and the elderly. As the only current specific treatment options for RSV are monoclonal antibodies, there is a need for efficacious antiviral treatments against RSV to be developed. We have previously shown that a group of synthetic non-coding single-stranded DNA oligonucleotides with lengths of 25–40 nucleotides can inhibit RSV infection in vitro and in vivo. Based on this, herein, we investigate whether naturally occurring single-stranded small non-coding RNA (sncRNA) fragments present in the airways have antiviral effects against RSV infection. From publicly available sequencing data, we selected sncRNA fragments such as YRNAs, tRNAs and rRNAs present in human bronchoalveolar lavage fluid (BALF) from healthy individuals. We utilized a GFP-expressing RSV to show that pre-treatment with the selected sncRNA fragments inhibited RSV infection in A549 cells in vitro. Furthermore, by using a flow cytometry-based binding assay, we demonstrate that these naturally occurring sncRNAs fragments inhibit viral infection most likely by binding to the RSV entry receptor nucleolin and thereby preventing the virus from binding to host cells, either directly or via steric hindrance. This finding highlights a new function of sncRNAs and displays the possibility of using naturally occurring sncRNAs as treatments against RSV.

## 1. Introduction

Respiratory syncytial virus (RSV) is one of the leading causes of acute lower respiratory tract infection (ALRI) in young children, immunocompromised individuals and the elderly. It has been estimated that, each year, approximately 34 million cases of RSV-related ALRIs occur in children under 5 years of age globally [1]. However, the COVID-19 pandemic may have temporarily changed this yearly incidence rate [2,3]. Furthermore, each year, this virus causes high morbidity and mortality rates in immunocompromised individuals, as well as in the elderly population [4]. Upon infection, no complete long-lasting immunological memory is created, leading to the occurrence of re-infections throughout one’s life, which highlights the importance of finding antiviral treatments against RSV [5,6,7]. There are several ongoing studies focusing on the development of efficacious antiviral treatments and vaccines against RSV. However, to date, the only treatment option for RSV is monoclonal antibodies, and they are primarily used in children at high risk of developing severe RSV-associated disease [8,9]. The infection route of RSV is not fully determined; however, studies have shown that RSV can infect cells via clathrin-mediated endocytosis [10]. Furthermore, the cell receptor nucleolin has been suggested to be an entry co-receptor for RSV [11,12].

We have previously shown that a 35-mer parent single-stranded DNA oligonucleotide (ssON) inhibited clathrin-mediated endocytosis (CME), which was dependent on the length of the ssON but largely independent from the sequence [13]. We found that extracellular oligonucleotides with lengths of 25–40 nucleotides were able to inhibit CME. As CME was reported to be an entry pathway for RSV, we previously investigated whether a 35-mer ssON could block RSV infection. We confirmed its high antiviral efficiency against RSV both in vitro and in vivo [14]. Interestingly, we discovered that the 35-mer ssON did not inhibit the endocytosis of the virus, as we had hypothesized, but rather that it blocked the virus from binding to the entry co-receptor nucleolin. As we previously established that synthetic non-coding single-stranded DNA oligonucleotides with lengths of 25–40 nucleotides can inhibit RSV infection, the aim of the present study was to investigate whether naturally occurring oligonucleotides within the length interval of 25–40 oligonucleotides exhibit the capacity to inhibit RSV. Therefore, in this study, we opted to evaluate whether small non-coding RNAs (sncRNAs) exhibit this effect, as they exist abundantly in vivo and have similar lengths to the parent 35-mer ssON.

SncRNAs are shorter than 200 nucleotides and often have a length of 18–30 nucleotides. SncRNAs are divided into several different subgroups such as microRNA (miRNA), small nucleolar RNA (snoRNA), transfer RNA (tRNA), ribosomal RNA (rRNA) and YRNA depending on their function and origin [15,16]. These have all been implicated in different regulatory roles, such as translation, DNA replication and RNA processing [17]. However, their roles during viral infections are not fully known, although it has been shown that the expression of host sncRNAs changes upon viral infection [15,18]. In this study, we evaluated whether naturally occurring sncRNAs, similar to 35-mer ssON, can inhibit RSV infection by interacting with the entry co-receptor nucleolin. For this purpose, we selected sncRNA fragments with a size of 30–40 nucleotides originating from YRNAs, rRNAs and tRNAs present in human bronchoalveolar lavage fluid (BALF) in healthy individuals [19]. By utilizing a GFP-expressing RSV A virus, we determined the infection rate of the virus in A549 cells in the presence of sncRNAs using flow cytometry. We show that all of the sncRNA fragments tested inhibited RSV infection, and that this inhibition of infection was likely due to interference with the viral binding to the host entry co-receptor nucleolin. This study highlights a new function of sncRNA fragments and creates the possibility of using naturally occurring sncRNAs as treatments against RSV infection.

## 2. Results

### 2.1. Naturally Occurring Small Non-Coding RNAs Inhibit RSV Infection In Vitro

We have previously shown that a set of synthetic single-stranded DNA oligonucleotides with a length between 25 and 40 nucleotides can inhibit RSV infection [14]. Based on this finding, here, we investigated whether naturally occurring non-coding RNAs exhibit a similar antiviral effect. Oligonucleotides over 20 nucleotides long are normally not taken up by cells without transfection or alternative intracellular delivery, which put delivery methods at the forefront for the antisense-based targeting of sequence located within cells [13]. Here, however, we added oligonucleotides to the extracellular space and investigated their tentative effects on the cell surface upon RSV infection.

We selected fragments of sncRNAs that were previously discovered in a study that compared the miRNA profiles of nanovesicles in the bronchoalveolar lavage (BALF) of asthmatic patients and healthy individuals because they also reported sequencing data of fragments present in BALF that were longer than miRNAs [19]. We re-analyzed the publicly available sequencing data from the BALF of the healthy controls and selected the most abundantly expressed 30–40-nucleotide-long transcripts for use in the present study. The selected sncRNAs were fragments originating from YRNAs, rRNAs and tRNAs from healthy individuals and here, we called them YRNA1, YRNA2, rRNA1, rRNA2, tRNA1 and tRNA2 (see Appendix A for sequences, annotations and descriptions). Hence, to the best of our knowledge, these selected RNA fragments were not evaluated functionally in any previous study.

As we utilized a GFP-expressing RSV, in which GFP is only expressed upon replication within cells, we were able to measure the inhibition of viral infection via the reduced expression of GFP. This method was validated by measuring viral content in the culture supernatants using RT-qPCR and by performing a viral transfer experiment followed by a TCID_50_ assay (Appendix A). The method validation data also showed that the 35-mer ssON treatment of cells resulted in reduced viral replication which was measured using RT-qPCR and established by the reduced transfer of infectious RSV, as determined using a TCID_50_ assay (Appendix A), which excluded the possibility that ssONs binds directly to GFP and shields the fluorescent signal.

In order to be able to evaluate sncRNA fragments in a 24 h RSV replication assay in vitro, it was necessary to introduce chemical stabilizations, as oligonucleotides of RNA or DNA origin are degraded by nucleases. Therefore, we added a 2′-O-methyl (2′OMe) to each base and a phosphorothioate (PS) backbone, as these modifications are commonly used for this type of oligonucleotides [20,21]. Furthermore, 2′OMe modification is a common naturally occurring modification of both rRNAs and tRNAs in vivo [22,23]. These modifications enhance the half-life of oligonucleotides by protecting them from degradation by nucleases.

A549 cells were pre-treated via the extracellular addition of the synthesized sncRNA fragments for 30 min prior to infection at a multiplicity of infection (MOI) of 1. The frequencies of viable infected cells were assessed using flow cytometry 24 h post infection. We used our parent ssON, which was a 35-nucleotide-long single-stranded DNA oligonucleotide, as a positive control, and a 15-nucleotide-long single-stranded RNA (15-mer RNA) as a negative control. We found that all of the sncRNA fragments, independent of origin, significantly inhibited RSV infection with an almost 90% reduction in vitro in A549 cells (Figure 1A), without compromising the viability of the cells, as assessed using a LIVE/DEAD^®^ Fixable near-IR Dead Cell Stain Kit and flow cytometry analysis (Figure 1B). The 15-mer RNA control did not inhibit the infection, confirming that the antiviral effect is dependent on the size of the oligonucleotide rather than a specific target sequence. Although the PS backbone modification enhanced the sncRNAs’ stability, nevertheless, we found that sncRNA fragments that contained the naturally occurring 2′OMe modification inhibited RSV infection. However, these sncRNAs had to be added every 5 min to the cell culture and in three times higher concentrations than the sncRNAs with both PS and 2′OMe, which were added only once (Appendix A). The addition of the sncRNA 2′OMe every 15 min had some effect on infection, but the effect was not as substantial as that seen when they were added every 5 min. This was most likely due to the degradation of the sncRNAs, as 2′OMe modifications do not protect oligonucleotides against nuclease degradation to the same extent as the combination of PS and 2′OMe modifications.

Next, we wanted to determine the efficiency of the sncRNA fragments against RSV infection by pre-treating A549 cells extracellularly with oligonucleotides using a concentration ranging from 500 nM to 0.5 nM. We found that all of the sncRNAs blocked the infection to varying degrees at low concentrations (0.5–50 nM). For example, the addition of YRNA1 and tRNA2 resulted in a nearly 50% reduction in infection frequency, whereas rRNA2 was less effective. However, at high concentrations (500 nM), all of the sncRNA fragments significantly blocked the RSV infection, with a reduction of almost 90% (Figure 2A–F).

We previously reported that 35-mer ssON has an IC_50_ of approximately 92 nM against RSV [14]. Here, we used non-linear regression analyses and determined that the majority of the sncRNA fragments had a similar IC_50_ to 35-mer ssON, ranging from 53 nM to 234 nM (Appendix A). In conclusion, we found that the YRNAs and tRNAs were the most efficient antiviral sncRNA fragments and inhibited RSV infection in a dose-dependent manner, while the rRNAs used were less effective against RSV infection.

### 2.2. Small Non-Coding RNAs Bind to the RSV Entry Co-Receptor Nucleolin

It has previously been shown that RSV infects cells by utilizing nucleolin as a co-receptor for viral entry [11,12]. We previously confirmed that RSV can bind nucleolin, and we also determined that ssONs bind to nucleolin [14]. Based on these results, we hypothesized that the ssON’s probable mode of action in vitro is to inhibit RSV infection by preventing the virus from binding to nucleolin.

Therefore, here, we investigated whether the sncRNA fragments, similarly to 35-mer ssON, could bind to nucleolin using a flow-cytometry-based nucleolin binding assay, as previously described [14,24]. A549 cells were kept at a low temperature and pre-treated with the sncRNA fragments to allow them to bind to the cell surface before the addition of an anti-nucleolin antibody. A secondary detection antibody was added, and the fluorescent signals of bound nucleolin antibody, in the presence or absence of the sncRNA fragments, were detected using flow cytometry. Similar to 35-mer ssON, we found that all of the tested sncRNA fragments bound to the RSV co-receptor nucleolin with a similar efficiency, as measured via both the significant reduction in the frequencies of anti-nucleolin-positive cells (Figure 3A, Appendix A) and via reduced mean fluorescence intensities (Figure 3B). The reduction in nucleolin antibody binding in the presence of 35-mer ssON or sncRNAs was confirmed using microscopy (Figure 3C–H, Appendix A). We included an isotype control and staining with the secondary antibody alone to ensure that our results were not due to the unspecific binding of the antibodies used. In summary, our results reveal that sncRNA fragments seem to exhibit the same mode of action as the parent 35-mer ssON against RSV infection in vitro in A549 cells by binding to the RSV entry co-receptor nucleolin.

To further evaluate whether the mode of action of 35-mer ssON and sncRNAs is via the steric hindrance of nucleolin rather than affecting the antiviral immune response, we measured whether 35-mer ssON influenced cytokine release in A549 cells. We found that the cells released IL-29 and IL-8 in response to RSV infection. However, 35-mer ssON did not enhance this antiviral response. On the contrary, there seemed to be reduced IL-29 and IL-8 production in the 35-mer ssON-treated cells, which was likely due to the lack of infection in the samples rather than a direct effect of ssON (Appendix A), in accordance with previously published data [25].

## 3. Discussion

Herein we show that naturally occurring non-coding sncRNA fragments found in the BALF of healthy humans can inhibit RSV infection in vitro in A549 cells. We determined that the fragments were efficient in their inhibition and had IC_50_ values varying from 53 nM to 234 nM. Furthermore, we found that all of the sncRNA fragments tested bound to or shielded the RSV entry co-receptor nucleolin, indicating that they are likely to compete with the virus to bind to nucleolin and/or prevent the virus from binding via steric hinderance.

We previously discovered that single-stranded DNA oligonucleotides with a size between 25 and 40 nucleotides can inhibit viral infections, such as RSV and influenza A [14,25]. However, single-stranded DNA fragments of this size are not common in vivo; therefore, we opted to test RNA fragments with the rationale that single-stranded sncRNAs exist in abundance in vivo. SncRNAs are non-coding RNAs that are shorter than 200 nucleotides, and they are divided into subgroups depending on their size, function and origin. Some of the most common sncRNAs that are found in BALF are rRNAs, tRNAs, and YRNAs, which exhibit different functions in regulating gene expression, cellular functions and immune responses [16,26,27].

It is well known that the abundance of host sncRNAs can be altered upon viral infections. For example, studies have shown that RSV can change the expression of host sncRNAs in A549 cells, Hep-2 cells and primary cells such as monocyte-derived dendritic cells (moDCs) [15,18]. If and how sncRNAs induced by infection affect the infection route of the virus is not well established, but recent studies have shown that RSV can utilize some host tRNA fragments induced by infection to increase its own replication [28]. However, the precise role of sncRNAs during RSV infection has yet to be determined.

Here, we demonstrate that the sncRNA fragments tested seemed to bind to or shield the host receptor nucleolin, which has been implicated as a co-receptor for RSV entry. A detailed entry pathway of RSV and the full role of nucleolin in this pathway has yet to be established. Some studies suggest that nucleolin is part of a multimeric complex responsible for entry, while a recent study suggested that nucleolin is recruited to the membrane upon RSV binding, indicating that nucleolin alone might not be sufficient for the binding or entry of RSV [12,29]. Furthermore, nucleolin is a multifunctional RNA-binding protein that has been implicated in the entry and infections of many viruses, making it a potential target for broad antiviral treatments [11,12,30]. Although our data indicate that sncRNA fragments bind to nucleolin, we cannot rule out the idea that they might bind to interaction partners in close proximity to nucleolin and thereby prevent the virus from binding to nucleolin via steric hindrance. Mapping the causality between the antiviral effect of sncRNAs to nucleolin and RSV infection has proven to be difficult, as knocking down nucleolin significantly reduces RSV infection, which confirms that nucleolin is indeed a receptor for RSV [11,31]. Furthermore, as nucleolin is an essential protein involved in many cellular processes such as the proliferation, transcriptional regulation, stability and transportation of mRNA, previous studies have shown that knocking out nucleolin is lethal to cells [32,33], precluding the possibility of conducting such experiments to show that the functional mechanism of ssONs is blocking nucleolin. As nucleolin is the only established functional receptor for RSV to date, it would be difficult to find other interaction partners for sncRNAs in relation to RSV infection. Although we propose that sncRNAs inhibit RSV infection by interacting with nucleolin, we cannot entirely exclude the idea that other binding partners and/or additional mechanisms are involved. Herein, we provide data suggesting that ssONs are unlikely to induce antiviral cytokine responses in A549 cells in vitro. However, we have previously shown that 35-mer ssON exhibit immune-modulatory properties during RSV infection in vivo, and it is possible, but remains to be proven, that sncRNA fragments also have immune-modulatory properties in a complex cellular environment during RSV infection in vivo.

The fragments used in this article were isolated from nanovesicles in human BALF; however, it is well established that sncRNA fragments are present extracellularly and that they utilize extracellular vesicles for transportation between cells [34,35]. The processing of sncRNA fragments from larger precursors in vivo is the consequence of stress-inducible ribonuclease activity, such as RNase 1, dicer (DCR1) and angiogenin (ANG). The modes of action differ, depending on the cellular location of the sncRNA fragmentation. For example, RNase 1 processes YRNA and tRNA fragments in the extracellular space, while ANG forms tRNA fragments intracellularly. These intracellular tRNA fragments (including tRF-Glu-CTC) are the result of ANG activation in airway epithelial cells upon RSV infection [36]. Although the exact molecular mechanisms involving sncRNA fragment formation, posttranscriptional regulation in host–virus interactions and immune responses are still to be clarified, sncRNA fragments may have valuable therapeutic potential.

The discovery that naturally occurring sncRNA fragments can bind to or shield nucleolin and inhibit RSV infection creates the possibility of using non-coding RNAs with sizes of 30–40 nucleotides as antiviral treatments. As these sncRNAs are naturally occurring in vivo, they may not evoke unwanted immune responses. Furthermore, the finding that these RSV-inhibiting sncRNA fragments are present in BALF, where the virus resides during infection, suggests that these fragments work as initial host defenses against viral infections and that they may limit the initial infection before the induction of innate immune responses. However, it is likely that relatively high fragment concentrations and/or the continuous output of RNA fragments are required for antiviral defenses, as RNAs are unstable in the extracellular environment. It is conceivable that chemical stabilizations would be required to enable the use of sncRNA fragments as antiviral treatments.

Altogether, our data demonstrate that naturally occurring small non-coding RNA fragments with sizes between 30 and 40 nucleotides can inhibit RSV infection in vitro. Furthermore, these fragments can interact with or shield the viral co-receptor nucleolin, suggesting that the mechanism of action in inhibiting infection in vitro is through competition with the virus for cellular binding or steric hindrance. This highlights a new function of sncRNAs and indicates the possibility of using sncRNAs as an antiviral agent against RSV infections.

## 4. Materials and Methods

### 4.1. Cells and Virus

A549 lung epithelial cells (obtained from and authenticated by ATCC) were cultured in complete DMEM high glucose (Hyclone, UT, USA) containing 5% FBS (Sigma Aldrich, MI, USA), 1% PEST (Hyclone, UT, USA), 1% L-glutamine (Hyclone, UT, USA), 1% Hepes (Hyclone, UT, USA) and 1% sodium pyruvate (Hyclone, UT, USA). Cells were used up to passage 50 and for each individual experiment, a different passage was used. Cell lines kept in the laboratory were regularly tested for mycoplasma. The day before infection, the cells were seeded in 24-well plates at a concentration of 5 × 10^4^ cells/well. Recombinant RSV expressing eGFP (rA2-eGFP) was rescued and produced as previously described [37].

### 4.2. Oligonucleotides

SncRNA fragments ranging from 30 to 40 nucleotides in size, present in healthy human BALF, were selected from sequencing data published by Francisco-Garcia et al. [19]. Specifically, we downloaded the Fastq files from the Gene Expression Omnibus website under accession number GSE152729. Sequences from all of the samples from healthy individuals were pooled together and pre-processed using the miRTrace software with the default options [38]. The most highly represented sequences were inspected manually using the NCBI nucleotide blast server (nr/nt database), and six sequences were selected to represent two of each of the tRNAs, rRNAs and YRNAs. These RNA oligonucleotides were then synthesized by Integrated DNA Technologies (IDT, Leuven, Belgium or IDT, Corallville, IA, USA) and contained a fully phosphorothioate (PS)-modified backbone and a 2′-O-methyl (2′OMe) modifications. The tRNA2 oligonucleotide was also produced with 2′OMe modifications alone. A 35-nucleotide-long single-stranded PS-modified DNA oligonucleotide was also produced by IDT. For sequences, see Appendix A.

### 4.3. RSV Infection and Analysis Using Flow Cytometry

Cells were treated with DNA or RNA oligonucleotides for 30 min by adding the oligonucleotides directly to the media, followed by RSV infection using a multiplicity of infection (MOI) of 1 for 2 h. Alternatively, cells were treated with tRNA 2′OMe (3 µM) for 15 min prior to infection with the new addition of tRNA 2′OMe (3 µM) every 15 min during the 45 min incubation with RSV, or cells were treated with tRNA 2′OMe (3 µM) for 5 min before infection and new tRNA 2′OMe (3 µM) was added every 5 min throughout the 45 min incubation with RSV. Cells were washed with PBS, resuspended in new media and incubated in 37 °C and 5% CO_2_. After 24 h of incubation, the cells were washed, trypsinized and stained with a LIVE/DEAD^®^ Fixable near-IR Dead Cell Stain Kit (Life Technologies, California, CA, USA). The data were acquired using a FACSVerse cytometer (BD, New Jersey, NJ, USA) and data analysis was performed using the FlowJo software (Tree Star).

### 4.4. Nucleolin Binding Assay

The nucleolin binding assay was performed as previously described with minor modifications [14,24]. Briefly, 2 mM EDTA was used to detach the cells and they were put on ice to cool. All of the treatments occurred at 4 °C to ensure that no cellular uptake could occur. The cells were treated with RNA or DNA oligonucleotides (1 µM) for 15 min prior to incubation with an anti-nucleolin antibody (Ab22758, Abcam, Cambridge, UK) for 30 min at 4 °C, at a concentration of 1 µg. Subsequently, the cells were stained with a secondary Alexa 488-conjugated donkey anti-rabbit IgG secondary antibody (5 µg/mL; Invitrogen, Massachusetts, MA, USA) for 30 min at 4 °C, followed by the acquisition of the samples using a FACSVerse cytometer (BD, New Jersey, NJ, USA). The analysis was performed using FlowJo software (Tree Star). Alternatively, after the secondary antibody staining, the cells were placed in imaging dishes and analyzed on a Zeiss Axio Observer Z1 microscope using a 20× objective followed by analysis using Zen Blue 3.2.

### 4.5. qPCR

The cells were treated with 35-mer ssON (1 µM) 2 h prior to RSV infection at MOI 0.1, 0.05 and 0.01. After 2 h, the cells were washed extensively before the new medium was added and the cells were incubated at 37 °C. Supernatants were collected 72 h post infection, and the viral RNA was extracted according to the manufacturer’s protocol using the QIAamp^®^ Viral RNA kit (Qiagen, Hilden, Germany). Viral load was assessed with RT-qPCR using the SuperScript^®^ III Platinum^®^ One-Step Quantitative RT-PCR System (Life Technologies, California, CA, USA). Known concentrations of the RSV-GFP stock were used as the internal standard. The following primers and probes were used, RSV A-forward: AGA TCA ACT TCT GTC ATC CAG CAA; RSV A-reverse: TTC TGC ACA TCA TAA TTA GGA G; RSV A_Probe: 6-FAM-CAC CAT CCA ACG GAG CAC AGG AGA T.

### 4.6. TCID_50_ Assay

The cells were treated with 35-mer ssON (1 µM) 2 h prior to RSV infection at MOI 0.1, 0.05 and 0.01. After 2 h, the cells were washed extensively before the new medium was added and the cells were incubated at 37 °C. Twenty-four hours post infection, the supernatant was collected, and a TCID_50_ assay was performed. A serial dilution of the supernatant was transferred to new A549 cells, and the infection was monitored 5 days post the transfer of the supernatant. To calculate the TCID_50_/mL, the dilution at which 50% of the wells showed GFP-positive signals was used in combination with the well-known Spearman–Karber method [39].

### 4.7. ELISA

A549 cells were treated with 35-mer ssON (1 µM) 2 h prior to RSV infection at MOI 1. After 2 h, the cells were washed extensively before the new medium was added and the cells were incubated at 37 °C. Supernatants were collected 24 h post infection, and the secretion levels of IL-8, IL-29 and INF-α were measured using ELISA according to the manufacturer’s protocol. All ELISA kits were acquired from Mabtech, Nacka, Sweden.

### 4.8. Statistical Analysis

GraphPad prism software version 6.07 was used to analyze all of the data. The Kruskal–Wallis one-way ANOVA test with Dunn’s multiple comparison was used when comparing the efficiency of different oligonucleotides. The different treatments were compared to the untreated infected RSV control. For all of the other experiments, the non-parametric Mann–Whitney test was used, comparing the oligonucleotide treatments to the untreated RSV or the nucleolin control. The IC_50_ values were obtained using non-linear regression analysis in Prism. *p*-value: ** *p* ≤ 0.01; *** *p* ≤ 0.001; **** *p* ≤ 0.0001. Lack of significance is not displayed in the figures. All the data are from three independent experiments in duplicates and presented as the mean ± standard error of the mean (SEM), unless otherwise stated in the figure legends.

## Figures and Tables

**Figure 1 ijms-23-05990-f001:**
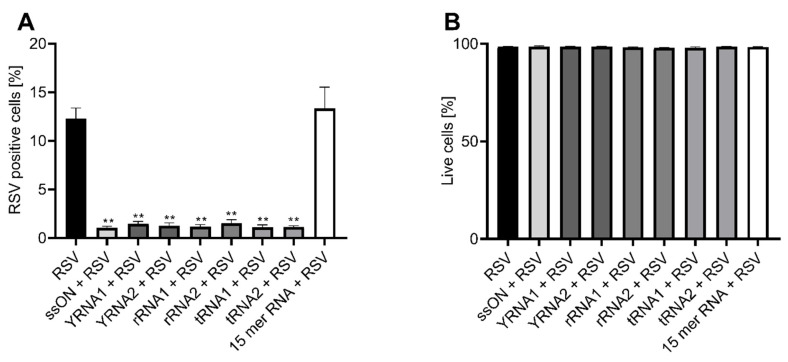
Naturally occurring small non-coding RNAs inhibit Respiratory syncytial virus (RSV) infection in vitro. A549 cells were treated with 1 µM of RNA or DNA oligonucleotides 30 min prior to infection with an RSV expressing GFP at MOI 1 for 2 h. Cells were subsequently washed with PBS, resuspended in media and further incubated for 24 h prior to staining with LIVE/DEAD^®^ Fixable near-IR Dead Cell Stain Kit. (**a**) The frequency of viable infected cells 24 h post infection and (**b**) the viability of cells treated with RSV and oligonucleotides were assessed using flow cytometry. Graphs depict data from three independent experiments conducted in duplicate. The data are presented as mean ± SEM, and significant differences were assessed using the non-parametric Mann–Whitney test by comparing all treatments to the RSV-infected control. *p*-value: ** *p* ≤ 0.01. Lack of significance is not displayed in the figure.

**Figure 2 ijms-23-05990-f002:**
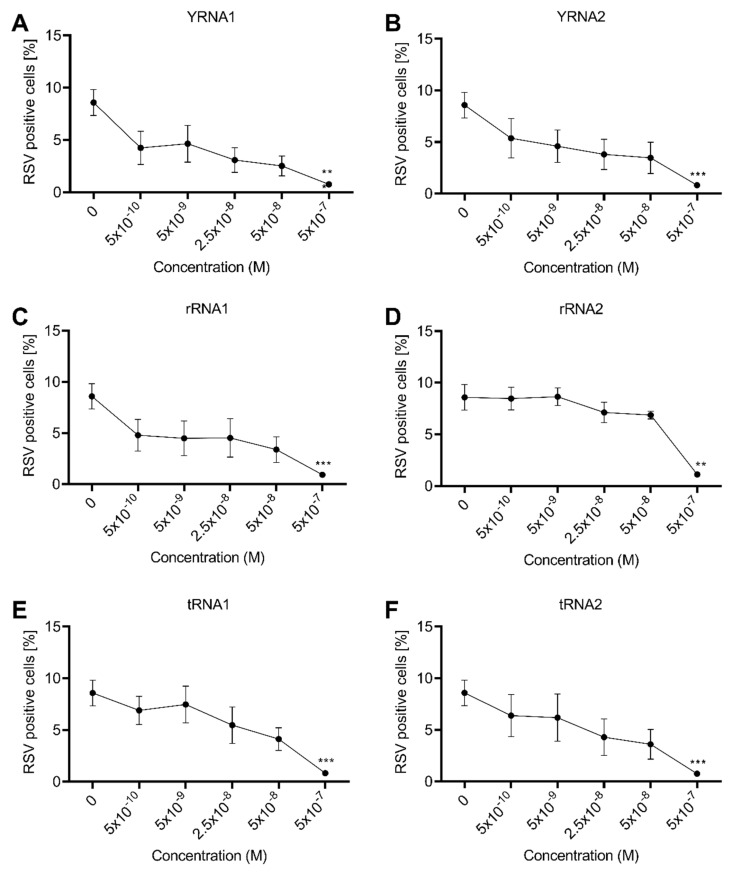
Small non-coding RNAs inhibit RSV infection in a concentration-dependent manner. A549 cells were treated with (**a**) YRNA1, (**b**) YRNA2, (**c**) rRNA1, (**d**) rRNA2, (**e**) tRNA1 or (**f**) tRNA2 at indicated concentrations for 30 min. Thereafter, cells were incubated with RSV at MOI 1 for 2 h. Subsequently, cells were washed, resuspended in media and incubated for 24 h. The recovered cells were stained with LIVE/DEAD^®^ Fixable near-IR Dead Cell Stain Kit and the proportion of viable infected cells was measured 24 h post infection using flow cytometry. The data are presented as mean ± SEM from three independent experiments conducted in duplicate. Significant differences were assessed using the Kruskal–Wallis one-way ANOVA test with Dunn’s multiple comparison, comparing each treatment to the RSV-infected control. *p*-value: ** *p* ≤ 0.01; *** *p* ≤ 0.001. Lack of significance is not displayed in the figure.

**Figure 3 ijms-23-05990-f003:**
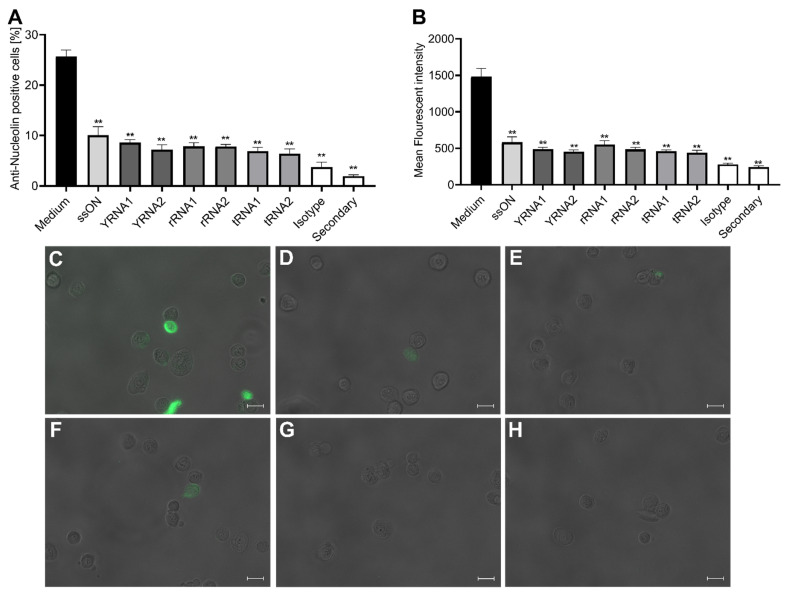
Small non-coding RNAs inhibit antibody binding to the RSV entry co-receptor nucleolin. A549 cells were treated with 35-mer ssON or RNA oligonucleotides (1 µM) for 15 min, followed by incubation with an anti-nucleolin antibody (1 µg/mL) for 30 min at 4 °C and then staining with an Alexa 488-conjugated donkey anti-rabbit IgG secondary antibody for 30 min at 4 °C. (**a**) The proportion of cells positive for the anti-nucleolin mAb and (**b**) mean fluorescence intensity (MFI) were measured using flow cytometry. The data are presented as mean ± SEM from three independent experiments conducted in duplicates. Samples containing staining with an isotype control and staining with the secondary antibody alone were included to ensure that no unspecific binding occurred. The nucleolin-positive gate was set according to the isotype control. Significant differences were assessed using the non-parametric Mann–Whitney test, comparing all treatments to the nucleolin control. *p*-value: ** *p* ≤ 0.01. Representative microscopy pictures showing nucleolin antibody staining in (**c**) baseline nucleolin control cells, (**d**) ssON-treated cells, (**e**) tRNA2-treated cells, (**f**) YRNA2-treated cells, (**g**) rRNA2-treated cells and (**h**) isotype-control-treated cells. Data representative from one experiment conducted in duplicate. Scale bar 20 µM.

## Data Availability

Not applicable.

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
