# Peer review of "Inhibition of Respiratory Syncytial Virus Infection by Small Non-Coding RNA Fragments"

_ijms, 2022, doi:10.3390/ijms23115990_

Round 1

Reviewer 1 Report

Dear authors,

Your topic is interesting.

It lacks a clear overview and the description of the structure of the paper at the end of the introduction.

Some improvements are required before reconsideration after major revision. In particular, a revision of the abstract is required since the second part [preparing...... road map] is completely unreadable and could mislead the reader on the real objective of this paper.

I think that this paper requires a few improvements as below:

  1. The abstract needs to get improved and in this case the first part of the abstract is actually a part that could be included in the paragraph of Introduction.
  2. It is necessary to highlight the novelty and contribution of the paper in the introduction.
  3. The introduction needs to improve, I am thinking about lacking theoretical background of the study. This should be located in the Introduction or in independent section.
  4. It is necessary to add some information about methodology and key results to make them clear.
  5. The discussion and conclusion part must be improved based on the academic paper.
  6. That would be good if you references also previous studies in the field.
  7. The results should be gathered independently so that authors‘ contribution is clear.

Reviewer 2 Report

This MS described that the inhibition of respiratory syncytial virus infection by small non-coding RNA fragments. This study was very straightforward and easy to be followed by viewers. It also provided an interesting idea about how to inhibit the replication of respiratory syncytial virus using small non-coding RNA fragments. Though, this study is indeed worthy a merit of publication. It is my opinion that there are still some points needed to be delineated and an extra experiment needed to be fulfilled  before it can be granted an acceptance without any preservation.

The specific comments were as follows:

  1. Line 126 and throughout the MS: Please leave a space within the letter of 500nM. It should be typed as 500 nM.
  2. Lines 222-225: Please compare the binding of the sncRNA with the nucleolin of cells treated with the anti-sense nucleolin with those without the anti-sense nucleolin treatment to clarify whether the sncRNA binds with the nucleolin or just its interaction partners. This is an important and feasible experiment which I advice the authors to execute.
  3. lines 291-296: It is absolutely necessary to give an explanation about why the authors examined the addition of the tRNA 2'OMe every 5 or 15 min during the incubation of cells with respiratory syncytial virus. For most readers, we need some background information as to why this issue has been selected to be tested.
  4. Figure 3 (A): Please give some descriptions about why the authors included the isotype and secondary groups in this study.  This will benefit the understanding of the readers.
